# Specific Phylotypes of *Saprolegnia parasitica* Associated with Atlantic Salmon Freshwater Aquaculture

**DOI:** 10.3390/jof10010057

**Published:** 2024-01-09

**Authors:** Kypher Varin Shreves, Marcia Saraiva, Tahmina Ruba, Claire Miller, E. Marian Scott, Debbie McLaggan, Pieter van West

**Affiliations:** 1International Centre for Aquaculture Research and Development (ICARD), Aberdeen Oomycete Laboratory, Institute of Medical Sciences, University of Aberdeen, Foresterhill, Aberdeen AB25 2ZD, UK; kypher.shreves1@abdn.ac.uk (K.V.S.); mrmsaraiva@gmail.com (M.S.); tahmina.ruba@bau.edu.bd (T.R.); d.mclaggan@abdn.ac.uk (D.M.); 2Department of Pathology, Faculty of Veterinary Science, Bangladesh Agricultural University, Mymensingh 2202, Bangladesh; 3School Mathematics and Statistics, University of Glasgow, Glasgow G12 8TA, UK; claire.miller@glasgow.ac.uk (C.M.); marian.scott@glasgow.ac.uk (E.M.S.)

**Keywords:** aquaculture, Atlantic salmon, epizootics, enzootics, fish, ITS, isolates, phylogenetics, saprolegniosis, *S. parasitica*

## Abstract

Saprolegniosis is a major destructive disease in freshwater aquaculture. The destructive economic impact of saprolegniosis on freshwater aquaculture necessitates further study on the range of *Saprolegnia* species within Atlantic salmon fish farms. This study undertook a thorough analysis of a total of 412 oomycete and fungal isolates that were successfully cultured and sequenced from 14 aquaculture sites in Scotland across a two-year sampling period. An ITS phylogenetic analysis of all isolates was performed according to whether they were isolated from fish or water samples and during enzootic or epizootic periods. Several genera of oomycetes were isolated from sampling sites, including *Achlya*, *Leptolegnia*, *Phytophthora*, and *Pythium*, but by far the most prevalent was *Saprolegnia*, accounting for 66% of all oomycetes isolated. An analysis of the ITS region of *Saprolegnia parasitica* showed five distinct phylotypes (S2–S6); S1 was not isolated from any site. Phylotype S2 was the most common and most widely distributed phylotype, being found at 12 of the 14 sampling sites. S2 was overwhelmingly sampled from fish (93.5%) and made up 91.1% of all *S. parasitica* phylotypes sampled during epizootics, as well as 67.2% of all *Saprolegnia*. This study indicates that a single phylotype may be responsible for *Saprolegnia* outbreaks in Atlantic salmon fish farms, and that water sampling and spore counts alone may be insufficient to predict *Saprolegnia* outbreaks in freshwater aquaculture.

## 1. Introduction

The total annual production losses in freshwater aquaculture due to saprolegniosis remain consistently high, with losses of 50% having been reported for over two decades, and with *Saprolegnia* responsible for at least 10% of all annual salmonid economic loss worldwide [1,2,3,4,5,6]. Past publications have identified species within the genus *Saprolegnia* as the cause of these infections [3,4,5,7,8,9,10]; however, the taxonomic classification of *Saprolegnia* spp., and oomycetes in general, has often been ambiguous.

The classical identification of *Saprolegnia* species was based upon the morphological characteristics of sexual structures, such as the oogonia, oospores, and antheridia [11,12,13]. Observations of such structures can often be difficult, particularly as the in vitro growth of many *Saprolegnia* isolates fails to produce these sexual structures. Following the introduction of molecular identification techniques, it became apparent that these methods could be ambiguous and misleading, resulting in the taxonomical misclassification of species (see [14] for a full list). The phylogeny of *Saprolegnia* has been re-evaluated and corrected [8,14], resulting in a robust model taxonomy which groups *Saprolegnia* species into 23 clusters based on genetic sequencing of the internal transcribed spacer (ITS) region of the nuclear ribosomal DNA (nrDNA) [14]. One finding of particular note was the re-classification of *S. hypogyna*, *S. parasitica*, and *S. salmonis*, (three species previously thought to be associated with saprolegniosis), into a single species, *S. parasitica* [8,14].

Saprolegniosis is caused by several species within the genus *Saprolegnia*, in particular, the species *S. australis*, *S. diclina*, and *S. parasitica* [3,5,7,8,9,10,14]; however, only a few published studies have investigated the range of *Saprolegnia* species within Atlantic salmon fish farms. Across farms in Chile and Canada, differences were reported in the species diversity of *Saprolegnia*, with *S. parasitica* making up the overall majority of isolated species [4,15]. Whether this may also be true of Scottish sites has not been determined. For *Saprolegnia* host–pathogen specificity, species-level identification is typically considered; however, it has been suggested that the interaction may actually be strain-specific [15]. Furthermore, whilst several studies have investigated *Saprolegnia* pathogenicity through the inference of molecular markers or through infection trials, this is the first study that has determined the range of *Saprolegnia* species present in aquaculture during epizootic and non-epizootic periods across several Scottish Atlantic salmon aquaculture sites. The distribution preference of species and strains between host and environment was also established; comparisons were also made between species isolated from fish epidermal tissue and environmental tank water. This work provides insight into the distribution profiles of *Saprolegnia* species that are present in Scottish freshwater aquaculture. The variation in *Saprolegnia* species abundance between water and fish suggests adaptive strategies to increase the probability of attachment to target hosts.

## 2. Materials and Methods

### 2.1. Sample Collection and Culturing

Samples from tank water and the epidermis of fish were taken on a monthly basis across fourteen Atlantic salmon freshwater aquaculture sites (anonymized as A to N). Sampling was performed by on-site personnel trained in proper sampling techniques and provided with all necessary sampling equipment.

Water samples were collected, whereby 1 mL of water from the sampling tank was placed into each well of a sterile 24 well plate. A single sterilized hemp seed was introduced to each well, acting as a growth medium for any oomycetes present in the water sample. The plates were sealed and sent by courier to the Aberdeen laboratory, where they were incubated at 12 °C. The plates were checked regularly over a 2-week period; any hemp seeds displaying signs of growth were transferred onto selective agar plates: Potato Dextrose Agar (60 mm plates) supplemented with antibiotics (0.1% (*w*/*v*) vancomycin (Alfa Aesar, Haverhill, MA, USA) and 0.5% (*w*/*v*) ampicillin (Sigma-Aldrich, St. Louis, MO, USA) and the anti-fungal compound 0.02% (*w*/*v*) pimaricin (Sigma-Alrich) (PDA-VAP).

To sample the epidermis of fish, two biopsies were taken from 9 fish per monthly sampling period. Fish from the same tank were selected randomly. The first biopsy was taken from behind the left pectoral fin; a 1 cm^2^ skin sample was transferred directly to 6-well PDA-VAP plates, skin down. The second biopsy comprised the left pectoral fin, transferred directly to 6-well PDA-VAP plates with the fin fanned out. The plates were sealed and sent by courier to the Aberdeen laboratory, where they were incubated at 12 °C. Purification was achieved through repeated sub-culturing by transferring agar plugs to fresh PDA-VAP plates. The purification of the cultures was confirmed by culturing in Pea Broth [16] with no antimicrobials at 24 °C. A clear broth after 24 h confirmed no bacterial contamination.

### 2.2. DNA Extraction

Pure cultures were grown in Pea Broth for ca. 48 h. DNA extraction was performed on collected, washed mycelia in accordance with the phenol/chloroform protocol described in [17]. Essentially, the mycelia were transferred to a 2 mL screw-cap tube with sterile glass beads. A volume of 800 µL of an extraction buffer (10 mM tris-HCI (pH 8), 50 mM EDTA, 0.5% (*w*/*v*) SDS, 0.5% (*v*/*v*) Triton X-100, and 0.5% (*v*/*v*) Tween 20) was added to the tube, with 2 µL of proteinase K (20 mg/mL, Promega, Madison, WI, USA). The samples were snap-frozen in liquid nitrogen, placed into a Fastprep24 5G (MP Biomedicals^TM^, Santa Ana, CA, USA), and run for four cycles of 45 s at 6.5 m/s to break open the cells by mechanical force. The samples were incubated at 55 °C for 30 min, with the tubes inverted every 10 min. The samples were cooled to room temperature (20 °C); then, 800 µL of equilibrated phenol/chloroform/isoamyl alcohol (25:24:1) (ACROS Organics^TM^, Geel, Belgium) was added. The tubes were vortexed and then centrifuged at 10,000× *g* for 10 min to separate the phases. Up to 1 mL of the upper phase was transferred to another 2 mL tube containing 1 mL of molecular-grade isopropanol (Fisher Scientific, Hampton, NH, USA) and incubated at −20 °C for at least 15 min. The samples were then centrifuged at 4 °C at 10,000× *g* for 10 min to precipitate genomic DNA. The supernatant was removed and the DNA pellet was washed with 70% (*v*/*v*) ethanol (molecular grade, Sigma Aldrich), centrifuged (1 min at 10,000× *g*), then washed with 100% molecular-grade ethanol, and air dried. The concentration and purity of the genomic DNA were assessed through Nanodrop and gel electrophoresis.

### 2.3. Species Identification

Taxonomic identification was performed by targeting the internal transcribed spacer (ITS) region using the primer set ITS_4_ALT (Forward: 5′-TCCTCCGCTTATTGATATG-3′) and ITS_5_ALT (Reverse: 5′-TGAAAAGTCGTAACAAGGTT-3′). The primers were selected for their universal binding, known to amplify a wide range of oomycetes and fungi [18,19]), as well as their large amplicon coverage, covering the ITS1, 5.8S rDNA, and ITS2 regions. The PCR reaction mix (25 µL per sample) contained a 1 µL DNA template (5 ng), 5 µL of 5X Phusion^®^ High Fidelity Buffer, 5 µL of MgCL2 (25 mM), 2 µL of dNTP’s (2.5 mM), 1 µL of ITS_5_ALT (10 µM), 1 µL of ITS_4_ALT (10 µM), 0.25 µL of Phusion^®^ High-Fidelity DNA Polymerase (0.5 U) (Thermo Scientific, Waltham, MA, USA), and 4.75 µL ddH_2_O. The PCR thermal cycle consisted of an initial denaturation of 30 s at 95 °C, followed by 35 cycles of 35 s at 95 °C, 15 s at 59 °C, 75 s at 72 °C, and a final step of 7 min at 72 °C. The PCR products were sequenced by Sanger sequencing (Eurofins, Luxembourg). The sequences were assessed and edited using Geneious Prime v2020.1.2 The chromatogram of each sequence was assessed for the quality of the reads, with any sequences that appeared to be contaminated or of low quality discarded from the dataset. In cases where both paired-end reads of a single sample were considered to be of good quality, both reads were aligned into a single consensus sequence using de novo assembly with the Geneious assembler at the highest sensitivity. In the event that assembly failed, the read with the highest quality was kept as the representative read for the sample. An initial trimming of both de novo assembled contigs and single reads was performed automatically, with an error probability limit of 0.01. The species identification of each sample was based on the results of the NCBI Basic Local Alignment Search Tool (BLAST) [20]. In the cases where isolates were assigned species-level assignment based on their blast results, the isolates were grouped based on their NCBI blast results using the most frequent nomenclature of the top 10–20 results. To ensure that the isolate sequences were then correctly grouped, an independent alignment was performed within each genus-level group. If it was found that an isolate assigned to a species did not closely match other sequences within the same assigned species group, then the sequence was re-assessed. Re-assessment typically resulted in the group being grouped into a species which had a much more similar sequence or assigned to an ‘sp’ group if no similar sequences could be found. A reference dataset attained from [14,15] was used for the resolution of the *S. parasitica* cluster, with a single reference sequence selected for each *S. parasitica* phylotype within the dataset (Appendix B Table A1). All alignments were performed within Geneious Prime v2020.1.2. using MUSCLE v2.5 [21] with the default settings and a maximum of 8 iterations. Consensus sequences were produced for all isolates with identical ITS sequences and used as a representative sequence for the respective species/phylotype. Alignment trimming was performed in Geneious Prime v2020.1.2.

### 2.4. Phylogenetic Analysis

The isolates were binned into water and fish based upon their sampling origin. The isolates (water and fish together) were also binned into epizootic and enzootic based on the output of a modified z-score model (Z3), which identified epizootic events from saprolegniosis mortalities (Appendix B, Figure A1). Phylogenetic trees were produced using TOPALi v.2.5 [22]. Model selection for each tree was performed using the model selection tool for Phy maximum likelihood (ML) and MrBayes v3.0. The optimal model was selected based on the resulting values of the Bayesian information criterion (BIC), with the lowest-value model selected. Phylogenetic relationships were assessed using both ML analysis with PhyML [23] and Bayesian inference with MrBayes [24]. Whilst model selection varied between trees, all ML models ran with 1000 bootstraps, and all MrBayes models were run with 5 × 10^6^ generations and a 35% burn-in. Treegraph2 (v2.15.0) [25] was used to visualize the ML trees and integrate support values from the Bayesian inference.

## 3. Results

### 3.1. Phylogenetic Analysis of All Isolates Sampled across Scottish Atlantic Salmon Aquaculture Sites

A total of 412 isolates were successfully cultured and sequenced from 14 aquaculture sites across the two-year sampling period (Appendix A, for complete isolate list); of those, 336 isolates were identified as oomycetes and 76 isolates were identified as true fungi. The 336 isolates were identified as belonging to the oomycete genera Saprolegnia (*n* = 228, 69.9%), Pythium (*n* = 95, 23%), Achlya (*n* = 10, 2.4%), Phytophthora (*n* = 2, 0.5%), and Leptolegnia (*n* = 1, 0.2%) (Figure 1). Saprolegnia was the most frequently isolated oomycete genus, followed by Pythium. The 228 Saprolegnia samples isolated consisted exclusively of five Saprolegnia species: *S. parasitica* (*n* = 151, 36.7%), *S. ferax* (*n* = 33, 8%), S. diclina (*n* = 20, 4.9%), *S. australis* (*n* = 14, 3.4%), and *S. delica* (*n* = 10, 2.4%), with *S. parasitica* being the most frequently sampled species overall. Of the 95 Pythium isolates sampled, *P. coloratum* was the most abundant (*n* = 24, 5.8%), followed by Pythium sp.5 (*n* = 21, 5.1%), and *P. flevoense* (*n* = 19, 4.6%). The oomycetes formed two distinct clusters, with Achlya, Leptolegnia, and Saprolegnia grouping in one, and Phytophthora and Pythium grouping in the other (Figure 1).

There was a clear differentiation between the fungal and oomycete isolates, although Mortierella formed a cluster distinct from other fungal samples that was marginally closer to the oomycete cluster (Figure 1). Despite the use of selective media containing pimaricin as an anti-fungal compound, 19% of the total isolates sampled were identified as true fungi (see Section 4).

### 3.2. Phylogenetic Analysis of Saprolegnia Genus

Isolates of the Saprolegnia genus were clearly grouped with their respective cluster, as designated by Sandoval-Sierra et al. [14] (Figure 2). All Saprolegnia isolates were grouped in a distinct phylogenetic branch. This branch consisted of *S. delica*, *S. australis*, *Saprolegnia* sp.1, *S. diclina*, *S. ferax*, *S. brachydanionis*, and *S. parasitica* (Figure 2), and almost all members of the branch were isolated from aquaculture sites in this study, with only two members, *Saprolegnia* sp.1 and *S. brachydanionis*, not isolated. *S. delica* and *S. ferax* displayed no variation in the ITS region. The site dispersal of *S. delica* and *S. ferax* was very narrow (Table 1), with 70% and 95% isolated from two different sites, respectively, both recirculating. Isolates of *S. australis* displayed enough variation in the ITS region to separate them into three different phylotypes, although the majority of *S. australis* isolates belonged to a single phylotype. The two more abundant phylotypes for *S. australis* were grouped with the reference sequences (Figure 2); the third phylotype had one isolate but no reference sequence, indicating that this phylotype is either rare, or the reference samples were simply not numerous enough to provide full phylotype diversity. *S. australis* was found at only two of the fourteen sites (Table 1), both recirculating systems. Isolates of *S. diclina* were clearly divided into two separate groups based on variation in the ITS, with the number of isolates in each group being twelve and eight (Figure 2). The reference sequences only represented one of these *S. diclina* groups; however, the NCBI BLAST results confirmed that the unrepresented group was not novel. *S. diclina* was sampled across just over half of the sites (eight of fourteen), with the total number of isolates sampled being relatively low (20) (Table 1).

The only species of *Saprolegnia* isolated across all sampling sites was *S. parasitica* (Table 1). The number of *S. parasitica* isolates varied greatly between sites, with site C sampling the highest number of all *S. parasitica* isolates (*n* = 51, 33.8%).

### 3.3. Phylogenetic Analysis of Saprolegnia Parasitica

This current analysis showed that the distribution of all 151 *S. parasitica* isolates were divided into five different phylotypes based on variations in the ITS (Figure 2). For clarity and consistency with the literature, the nomenclature used for the phylotypes in this study is adopted from the study of Sarowar et al. [15], who identified four *S. parasitica* phylotypes/strains using ITS and Cox1 regions and designated them as S1–S4. S1 was not isolated from any Scottish fish farm; the five isolated phylotypes were designated as S2–S6 (Table 2).

The resolution of the *S. parasitica* phenotypes isolated was achieved using eight reference sequences (four from [14] and four from [15]). The Scottish isolates were grouped with the five phylotypes as designated [14,15] (Appendix B, Figure A2).

*S. parasitica* phylotype S2 was by far the most abundant phylotype sampled (70.9%) on Scottish fish farms, and it was also the most widely distributed *S. parasitica* phylotype, sampled across all but two sites. The second most abundant *S. parasitica* phylotype was S6 (23.8%) (Table 2). S6 was isolated from only three sites, one recirculating site and two sites geographically isolated from the mainland UK.

### 3.4. Analysis of Nucleotide Changes of S. parasitica Phylotypes

The ITS consensus sequences from each isolated *S. parasitica* phylotype (S2–S6), and also S1 from Sarowar et al. [15], were compared with the reference sequence KF717839.1 (SAP0208) [14]. The nucleotides are numbered in accordance with this reference sequence. Differences are seen at 10 different bp positions (Figure 3).

### 3.5. Phylogenetic Comparison of Saprolegnia between Water and Fish Isolates

When *Saprolegnia* isolates were grouped into those sampled from the epidermis of Atlantic salmon and those isolated from tank water, more than three times the number of isolates were sampled from fish (*n* = 172, 75.4%) than from water (*n* = 56, 25.6%) overall. The isolates in fish and in water were *S. australis*, *S. diclina*, and *S. delica* (Table 3), whereas the number of *S. ferax* isolated from fish (*n* = 23) was more than double the number found in water (*n* = 10). The largest difference in the abundance of water and fish isolates was observed with *S. parasitica*, with the vast majority of *S. parasitica* isolates sampled from fish (*n* = 130, 86.1%). S2, the most abundant *S. parasitica* phylotype, was overwhelmingly sampled from fish (*n* = 100), with only seven samples isolated from water (Table 3). The second most abundant phylotype, S6, had 23 isolates sampled from fish and 13 isolates sampled from water.

### 3.6. Phylogenetic Comparison of Saprolegnia between Enzootic and Epizootic Periods

At the time of sampling, 225 of a total of 228 *Saprolegnia* isolates could be divided based upon their epizootic status. It was found that the majority (164 isolates (78.5%)) were sampled during enzootic periods, rather than epizootic periods. The only *S. parasitica* phylotype sampled during both enzootic and epizootic periods was *S. parasitica* phylotype S2. Phylotypes S4 and S6 were found exclusively during enzootic periods, and S3 and S5 were found exclusively during epizootic periods (Table 4). During epizootic periods, almost all *S. parasitica* isolates were S2 (91.1%), with the remainder being S5 (6.7%) and S3 (2.2%). Both phylotypes of *S. diclina* were isolated during enzootic (*n* = 16) and epizootic (*n* = 4) periods (Table 4). *S. delica* and *S. australis* were also found during both enzootic and epizootic periods; the numbers were low and relatively similar for both periods, whereas two of the *S. australis* phylotypes were only found during epizootic periods. In contrast, *S. ferax* was exclusively found during enzootic periods (Table 4).

This was also the case for *S. parasitica* S6; it was the second most frequently sampled *S. parasitica* phylotype, and was isolated exclusively during enzootic periods, as was S4, although the numbers for S4 were low (Table 4).

## 4. Discussion

This study has demonstrated that *Saprolegnia* is ubiquitous across Scottish freshwater Atlantic salmon aquaculture, with the majority of *Saprolegnia* isolated during enzootic periods. This study also showed that, at the species level, *S. parasitica* is the primary *Saprolegnia* species sampled during saprolegniosis epizootics.

Phylogenetically, all *Saprolegnia* isolates were grouped within the species-level clusters as described by Sandoval-Sierra et al. [14]. In Chile, Sandoval-Sierra et al. [4] reported the isolation of all *Saprolegnia* species identified in this study, in addition to *Saprolegnia* sp.1 and *Saprolegnia* sp.2. In contrast, in Nova Scotia, no isolates of *S. australis*, *S. delica*, or *S. diclina* were identified, although the addition of *S. torulosa* was reported [15]. *S. parasitica* and *S. ferax* were the only species consistently found within Atlantic salmon aquaculture across all three studies. Although there are some commonalities, it seems that *Saprolegnia* species richness is not uniform within Atlantic salmon aquaculture at a global or even national level, but instead may be unique to individual aquaculture sites.

In the current study, the majority of isolates of *S. australis*, *S. delica*, and *S. ferax* were found at two sites, and these sites were both recirculating sites. These sites recycle most of their water and have greater biocontrol in comparison to flow-through sites. Recirculating sites are designed to reduce the introduction of pathogenic microorganisms, with, for example, controlled decontamination entrances and extensive water treatment facilities. However, it is possible that such infrastructure also prevents microorganisms from being removed. If treatments inhibit, rather than kill, *Saprolegnia*, zoospores may encyst and reside in biofilms [26]. Environmental conditions such as temperature are typically more stable within recirculating sites than flow-through sites; this stability may be more favorable for some species of *Saprolegnia.* The extensive water treatment found within recirculating sites may also reduce the ability of other microorganisms to survive which compete with or inhibit the growth of specific *Saprolegnia* species, which may partially explain why species such as *S. australis*, *S. delica*, and *S. ferax* are almost exclusively found in recirculating sites and in such abundance.

*S. parasitica* was by far the most abundant *Saprolegnia* species isolated, and the only species to be found ubiquitously across 12 out of 14 Scottish aquaculture sites tested. In Atlantic salmon studies, an increased abundance of *S. parasitica*, relative to other species of *Saprolegnia*, is a common phenomenon seen across different geographic locations, in both wild and captive populations [4,15,27,28]. This pattern may suggest a strong host–pathogen inter-relationship, with *S. parasitica* adapting to host environmental conditions and proliferating at a much greater rate than other *Saprolegnia* species. Studies by Matthews [28] found evidence of high genetic variation among *S. parasitica* isolates, indicating their adaptive potential.

The phylogenetic analysis of all *S. parasitica* species isolated in this study showed five distinct phylotypes, more phylotypes than any other species of *Saprolegnia* isolated. No isolates were obtained of phylotype S1. Phylotype S2 was the most common and widely distributed phylotype, being found at 12 of the 14 sampling sites. Isolates with ITS regions matching the *S. parasitica* S2 ITS phylotype seem to be the most widely reported *S. parasitica* phylotype worldwide. *S. parasitica* isolates that can be designated as S2 are found in Canada, Chile, Croatia, Egypt, Norway, Poland, Scotland, Spain, and Switzerland [4,14,15,29,30,31,32] (Table 5). In a study of oomycete isolates from trout farms in Croatia, 86% of all *S. parasitica* isolates could be designated as S2, with a single isolate of S6 sampled from water [29]. During an outbreak of saprolegniosis in Nile tilapia in Egypt, *Saprolegnia* spp. were isolated [30]. Two out of six isolates of *S. parasitica* recovered from naturally infected *O. niloticus* could be designated as the S2 phylotype based on ITS sequencing (Table 5) [4,15]. In the study by Ravasi et al. [32], isolates were collected from different locations in Switzerland, including fish hatcheries, fish farms, rivers, and lakes. The isolates were categorized using a multilocus sequence typing (MLST) scheme, using seven housekeeping genes; a diploid sequence type (DST) was assigned to each unique combination of alleles. This allowed a higher resolution of phylotyping, and there were a total of 10 different DSTs, with DST3 being the most common [32]. Based on ITS sequencing, we could assign eight of the ten DSTs to the S2 phylotype. The other two DSTs could be assigned to the S1 phylotype. Thus, from studies reported to date, it seems that a single phylotype dominates other phylotypes in salmonid aquaculture. The S2 ITS sequence displayed a 100% match with the *S. parasitica* strain CBS 223.65 (C65), a strain originally sampled from Northern pike (*Esox lucius*) in 1965, and also *S. parasitica* N12 (VI02736), originally sampled from the parr of Atlantic salmon in Scotland in 2002 [33,34], both of which are widely used as model strains for *S. parasitica* studies.

The S6 ITS phylotype was the second most frequently isolated phylotype of *S. parasitica* found at three of the fourteen sampling sites. The highest frequency of S6 was found at a recirculating site; this site had relatively high numbers of both *S. ferax* and *S. parasitica*, with *S. parasitica* phylotypes S6 and S2 being relatively equal in numbers. The other two sites that S6 was isolated from are unique in this study, as they are separated from mainland Scotland, and only *S. parasitica* S6 was isolated from these sites. The separation of land masses often creates geographic isolation in terrestrial species, as the water between the land masses acts as a barrier for migration. Although reports of the S6 phylotype are much less common than those of S2, isolates have been found in Chile, Ecuador, Poland, Russia, Spain, Scotland [4,8,14,31,33], and Korea [35], and a single isolate was found in Croatia [29] (Table 5). A recent report of the diversity and distribution of culturable Saprolegniaceae species in freshwater ecosystems in Korea obtained a total of nine *Saprolegnia* strains [35]. There were only four isolates of *Saprolegnia parasitica* isolates found, all of which could be designated as the S6 phylotype based on the ITS sequence analysis.

*S. parasitica* phylotypes S3, S4, and S5 were isolated far less frequently than S2 or S6, suggesting that they may be less well adapted to an aquaculture environment. The S3 phylotype was found at two sites and has only been previously reported in Canada [15], though an unpublished submission on GenBank has also identified the phylotype in Scotland (accession number: MW356889.1). The S4 phylotype was found only at one site. From the isolates reported in GenBank, S4 has been found in Argentina, Canada, and the USA [14,15,36]. The S5 phylotype was isolated from only two of the fourteen sampling sites, both flow-through sites. From the isolates reported on GenBank, the S5 phylotype is found in the Czech Republic, Ecuador, Spain, and the USA [14,36,37].

**Table 5 jof-10-00057-t005:** Global distribution of *S. parasitica* phylotypes S1–S6 designated by 600 bp ITS region. The GenBank comparison of ITS sequences illustrates the different countries in which phylotypes S1–S6 are found. The relative numbers of isolates are not depicted but discussed in text.

*S. parasitica* Isolate	Country of Origin	Host/Habitat	ITS Phylotype	GenBank	Ref.
SAP1091	Argentina	River water	S4	KF717864	[14]
Isolate #5	Canada	*Salmo salar* ^1^	S1	MK849947	[15]
Isolate #4	Canada	*Salmo salar* ^2^	S2	MK849946	[15]
Isolate #49	Canada	*Salvelinus fontinalis*	S3	MK849963	[15]
Isolate #1	Canada	Water	S4	MK849943	[15]
Li16	Czech Republic	*Astacus astacus*	S5	KF386710	[37]
SAP0522	Chile	*Salmo salar* ^3^	S2	KF717845	[4]
SAP0530	Chile	*Salmo salar*	S6	KM095949	[4]
Isolate B1L1	Croatia	*Oncorhynchus mykiss* ^4^	S2	MT555893	[29]
Isolate 122	Croatia	Water	S6	MT555889	[29]
SAP1230	Ecuador	River water	S5	KF717872	[14]
SAP1381	Ecuador	River water	S6	KF717876	[14]
SA221013	Egypt	*Oreochromis niloticus*	S2	ON797303	[30]
W9	Korea	Water	S6	ON075413	[35]
CBS 223.65	Netherlands	*Esox lucius*	S2	KF717879	[33]
SAP1484	Norway	*Salmo salar*	S2	KF717880	[14]
SAP0254	Poland	Water	S2	KF717840	[14]
SAP0257	Poland	Water	S6	KF717842	[14]
CBS 113187	Russia	N/A	S6	HQ644005	[33]
DD.37.04	Scotland	*Salmo salar*	S2	OQ678594	This study
VI0 5977	Scotland	*Salmo salar* eggs	S2	HG329736	[31]
N12 (VI-02736)	Scotland	*Salmo salar*	S2	NCBI: txid983306	[34]
AA.35.02.04	Scotland	Aquaculture tank water	S3	OQ678418	This study
DD.57.10	Scotland	*Salmo salar*	S4	OQ678598	This study
GG.48.02	Scotland	*Salmo salar*	S5	OQ678653	This study
EE.19.06	Scotland	*Salmo salar*	S6	OQ678612	This study
VI0 6009	Scotland	*Salmo salar* eggs	S6	HG329739	[31]
SAP0601	Spain	Water	S2	KF717849	[14]
SAP1203	Spain	Water	S5	KF717870	[14]
SAP26	Spain	Water	S6	AM228725	[8]
S026	Switzerland	*Salmo trutta* ^5^	S1	MH030519	[32]
S001	Switzerland	*Salmo trutta* ^6^	S2	MH030499	[32]
UNCW314	USA	N/A	S4	DQ353545	[36]
UNCW373	USA	N/A	S5	DQ393557	[36]

^1^ S1 also isolated from brown trout (*Salmo trutta*), brook trout (*Salvelinus fontinalis)*, Arctic charr (*Salvelinus alpinus)*, and striped bass (*Salvelinus alpinus*). ^2^ S2 also isolated from brook trout (*S. fontinalis)*, striped bass (*S. alpinus*), and eggs of Atlantic salmon (*Salmo salar*) and brook trout (*S. fontinalis).*
^3^ S2 also isolated from rainbow trout (*Oncorhynchus mykiss*) and from water. ^4^ S2 also isolated from brown trout (*S. trutta*), and rainbow trout (*O. mykiss*) eggs. ^5^ S2 also isolated from common minnow (*Phoxinus phoxinus*), grayling (*Thymallus thymallus*), whitefish (*Coregonus clupeaformis*), and Atlantic salmon (*Salmo salar*). ^6^ S2 also isolated from marble trout (*Salmo marmoratus*).

Regarding the distribution of isolates between water and fish, 75% of all *Saprolegnia* isolates were sampled from the epidermis of Atlantic salmon, with the remaining from tank water. The numbers of *S. delica*, *S. diclina*, and *S. australis*, isolates were relatively even between water and fish, whereas a higher number of *S. ferax* was isolated from fish (70%). The greatest difference between water and fish was seen for the *S. parasitica* isolates. Most notable was the distribution of the *S. parasitica* S2 phylotype, with 93.5% of the 107 isolates being sampled from fish; thus, during epizootics, virtually no *S. parasitica* S2 was detected in water samples, indicating that assessing the risk of saprolegniosis by sampling water for *Saprolegnia* or *Saprolegnia* zoospores would not provide a true reflection of saprolegniosis risk.

When an analysis was carried out on the distribution of isolates during enzootic and epizootic periods, all species of *Saprolegnia* isolated in this study were found during both enzootic and epizootic periods, with the exception of *S. ferax*, which was found only during enzootic periods. Although *S. ferax* is a known pathogenic species, it is primarily pathogenic to amphibians [38]. *S. ferax* has been isolated from infected Atlantic salmon at aquaculture sites in both Chile and Canada [4,15]; however, cases were rare (5.6%, and 7.1% of isolates, respectively), with both studies concluding that *S. ferax* was not a primary infector. The apparent absence of *S. ferax* during epizootics was not anticipated, as it was the second most frequently sampled *Saprolegnia* species and was isolated more often from fish than from water. However, a caveat to this observation is that all *S. ferax* samples were isolated almost exclusively from a single site (97%), which was a recirculating site. *S. australis* and *S. delica* were low in number and relatively evenly distributed between enzootic and epizootic periods. *S. diclina* was primarily found during enzootic periods (80%). The majority of *Saprolegnia* isolates sampled during epizootics were identified as *S. parasitica* (73.8%), which was anticipated, as *S. parasitica* is known as the primary cause of saprolegniosis in salmon [5]. S6 was never isolated during an epizootic period, supporting suggestions that pathogenicity varies between isolates of *S. parasitica* [39,40]. It is important to note that, whilst the highest number of isolates of *S. ferax* and *S. parasitica* phylotypes S2 and S6 were sampled from site C than any other site, the isolates found across the five epizootics that occurred at site C during the sampling period were exclusively S2. Overall, the S2 phylotype made up 91.1% of all *S. parasitica* phylotypes sampled during epizootics in Scotland, 97.3% of all *S. parasitica* strains sampled from infected Atlantic salmon in Chile [4], 63.2% of all *S. parasitica* strains sampled from infected Atlantic salmon in Canada [15], and 86% of all *S. parasitica* strains sampled from infected trout in trout farms in Croatia [29].

In addition to *Saprolegnia*, several other genera of oomycetes were isolated from the sampling sites, including *Achlya*, *Leptolegnia*, *Phytophthora*, and *Pythium*. Both *Leptolegnia* and *Phytophthora* are known to contain highly pathogenic species [6,41,42]; however, neither are associated with Atlantic salmon, and the infrequence with which these isolates were obtained in this study indicated their low abundance. A greater species richness of *Achlya* was present across farm sites, relative to *Leptolegnia* and *Phytophthora*. Whilst infection of *Achlya* species has been reported in several fish species [6,43,44], there are no reports of *Achlya* infection in Atlantic salmon. *Pythium* was the second most frequently isolated genera of oomycete, after *Saprolegnia*. Of the three most abundantly isolated *Pythium* species, only *P. coloratum* and *P. flevoense* could be identified with confidence via NCBI BLAST. Whilst many species of *Pythium* are pathogenic, the majority of pathogenic species are plant pathogens, with few species known to infect animals [6,45,46,47,48]. *P. coloratum* is associated with the infection of carrots [49], and *P. flevoense* has been associated with infection in Ayu fish larvae (*Plecoglossus altivelis*) [50]. *Pythium* species have been isolated from fish; however, very few studies have associated *Pythium* with infections [45,47,50,51], and no study to date has associated *Pythium* with infection in Atlantic salmon. *Pythium* are ubiquitous, found in freshwater worldwide [6,48]. Considering the lack of association with fish infection, the abundance and richness of *Pythium* species found in aquaculture sites may be influenced by their respective water sources, and it is likely that they have no notable effect on fish. However, an indirect effect of *Pythium* on the manifestation of saprolegniosis cannot be ruled out.

We also isolated 76 fungal isolates contributing to 19% of the total isolates obtained during this study. The true fungi primarily belonged to the genus Mortierella (*n* = 64, 15.5%), with Mortierella hyalina being the most frequently isolated fungal species (*n* = 24, 5.8%). Despite the use of selective media containing pimaricin, an anti-fungal compound, the most common fungal isolates belonged to the genus Mortierella, which is consistent with the finding of past studies that have shown reduced inhibitory effects of pimaricin on the growth of Mortierella [52,53,54]. Furthermore, some Mortierella produce white, cotton-like colonies, which are very similar in appearance to oomycetes such as Pythium and Saprolegnia [55,56,57]. It is likely that the anti-fungal resistance and physical appearance of Mortierella caused a bias during sample selection, resulting in an overabundance of these isolates. For information, we chose to list all isolates found under the sampling regime used; however, no further interpretation regarding true fungi abundance can be made.

## 5. Conclusions

This study established the diversity of *Saprolegnia* species within Scottish freshwater Atlantic salmon aquaculture. The variation in *Saprolegnia* species abundance between water and fish suggest adaptive strategies to increase the probability of attachment to target hosts. This study reports that of the current six different *S. parasitica* phylotypes in the literature, phylotype S1 was not detected in Scottish freshwater aquaculture; phylotypes S3, S4, and S5 were all very low in abundance. Phylotype S6 was the second most abundant, but by far the most ubiquitous across the farm sites was the *S. parasitica* S2 phylotype, present in high abundance, particularly on fish, suggesting that this particular phylotype is the primary cause of Atlantic salmon saprolegniosis, with similar observations found in both Canada and Chile.

## Figures and Tables

**Figure 1 jof-10-00057-f001:**
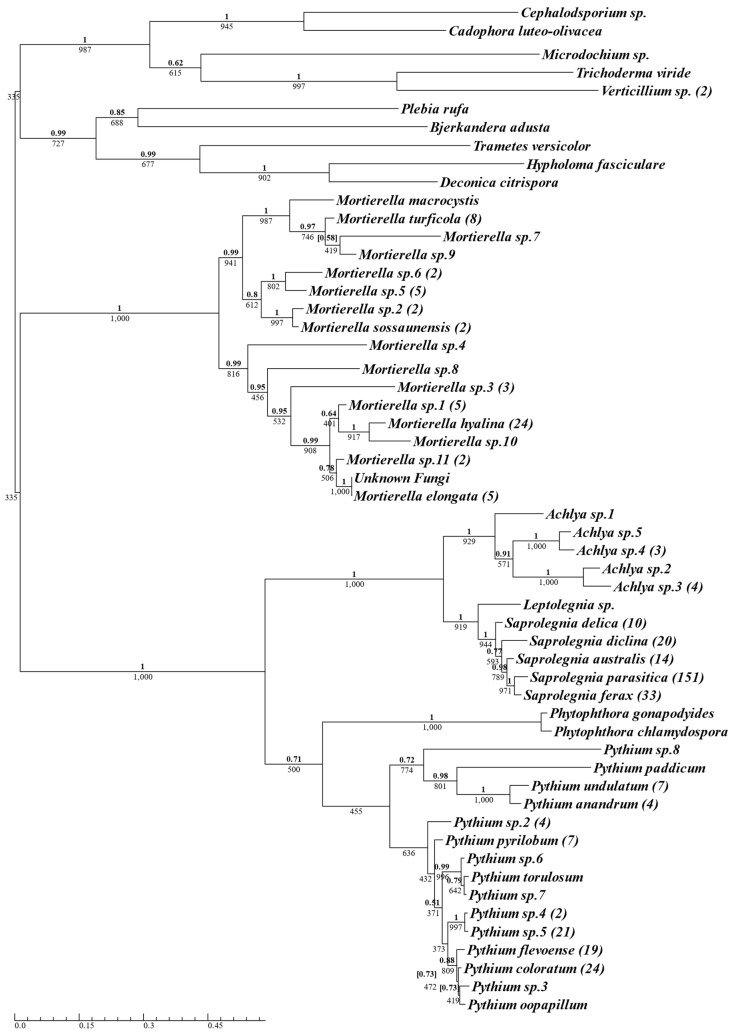
Phylogenetic summary of all oomycete and fungal species isolated using oomycete-selective media, across Scottish Atlantic salmon aquaculture sites. A single consensus ITS sequence was used for isolates belonging to the same species, with the number of isolates for each species in brackets next to the species name. Phy maximum-likelihood tree was produced using the TrN + I + G model with Bayesian inference using the GTR + I + G model. Supporting values for each branch are displayed with maximum-likelihood bootstrap values below and Bayesian support values above (bold). Conflicting Bayesian topologies are indicated with posterior probabilities in square brackets.

**Figure 2 jof-10-00057-f002:**
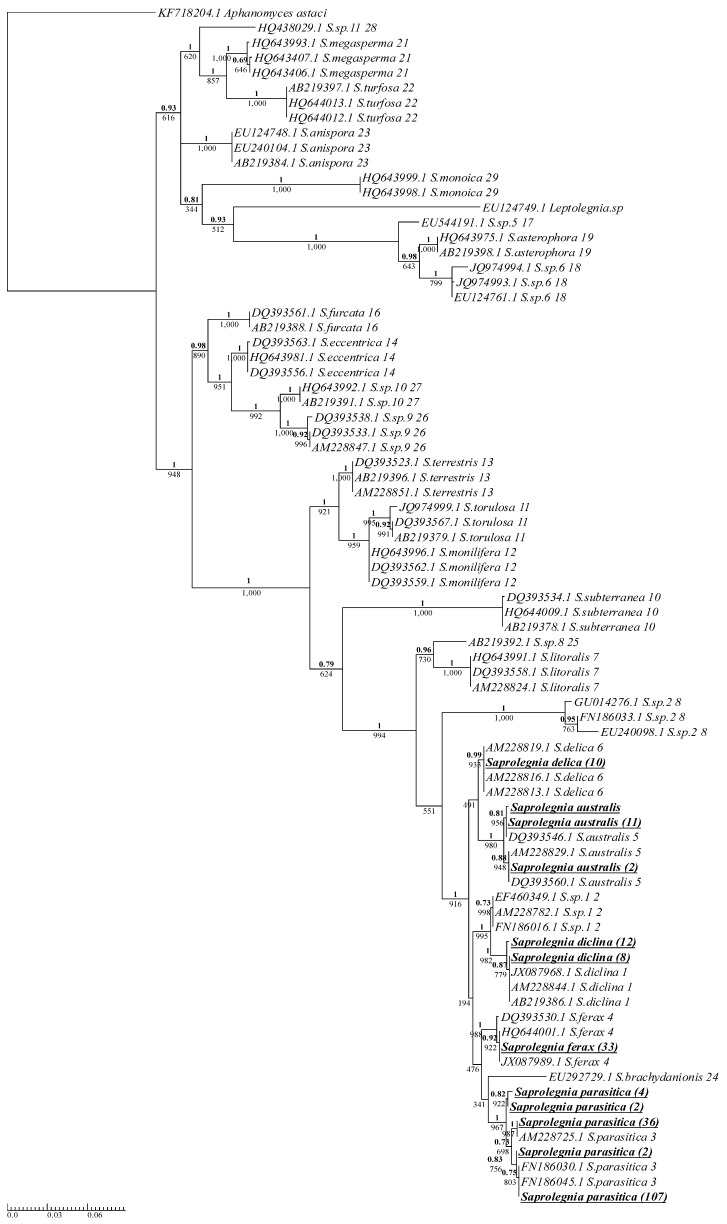
Phylogenetic summary of all Saprolegnia species isolated across Scottish Atlantic salmon aquaculture sites alongside model reference sequences (Appendix B, Table A1). A single consensus sequence was used for isolates with identical ITS sequences, with the number of isolates for each consensus in brackets next to the species name. Isolates from this study are boldened and underlined. Reference sequences were taken from [14] and are listed with their NCBI accession number, species names, and cluster number as designated by the reference study. Phy maximum-likelihood tree and Bayesian inference were both produced using the GTR + G model. Supporting values for each branch are displayed with maximum-likelihood bootstrap values below and Bayesian support values above (bold).

**Figure 3 jof-10-00057-f003:**
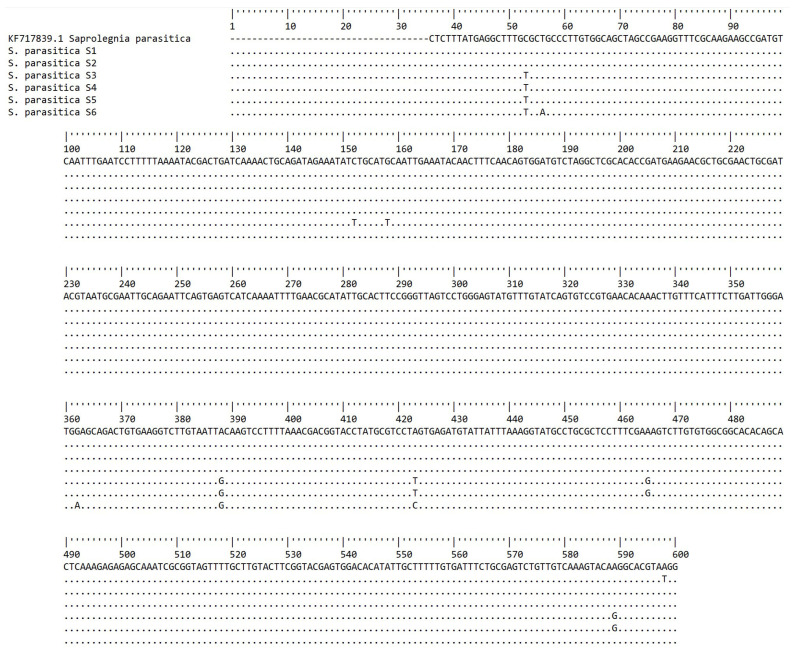
Nucleotide sequences of the ITS region for *S. parasitica* phylotypes. Nucleotides are numbered in accordance with the reference sequence KF717839.1 [14]. Nucleotides identical to the reference sequence for each position are represented by dots; the nucleotides which deviate from the reference are shown. Deviations are seen at the 53, 56, 152, 158, 362, 388, 423, 465, 589, and 598 bp positions.

**Table 1 jof-10-00057-t001:** Number of isolates of each *Saprolegnia* species obtained from Atlantic salmon farms in Scotland. Isolates were identified to species level via NCBI BLAST and confirmed through comparison to model sequences identified via molecular operational taxonomic units [14].

	Aquaculture Site														
*Saprolegnia* Species	A	B	C	D	E	F	G	H	I	J	K	L	M	N	Total
*S. australis*	0	0	0	0	0	0	0	0	0	12	0	0	0	2	14
*S. delica*	1	0	0	0	0	0	0	0	1	7	0	0	1	0	10
*S. diclina*	3	2	0	0	4	0	3	2	4	1	0	0	0	1	20
*S. ferax*	0	0	32	0	0	0	0	0	0	0	0	0	1	0	33
*S. parasitica*	3	13	51	6	2	7	19	7	13	1	1	9	11	8	151

**Table 2 jof-10-00057-t002:** *Saprolegnia* species phylotypes obtained from Atlantic salmon farms in Scotland. *S. parasitica* phylotypes were identified via MUSCLE alignment and confirmed via comparison with model sequences published in [14]; S1 was not isolated from any Scottish fish farm.

	Aquaculture Site														
*Saprolegnia* Phylotypes	A	B	C	D	E	F	G	H	I	J	K	L	M	N	Total
S2	1	13	24	4	0	0	15	7	13	1	1	9	11	8	107
S3	1	0	0	0	0	0	1	0	0	0	0	0	0	0	2
S4	0	0	0	2	0	0	0	0	0	0	0	0	0	0	2
S5	1	0	0	0	0	0	3	0	0	0	0	0	0	0	4
S6	0	0	27	0	2	7	0	0	0	0	0	0	0	0	36

**Table 3 jof-10-00057-t003:** Number of isolates of each *Saprolegnia* species obtained from water and from fish in Atlantic salmon farms in Scotland. A phylogenetic analysis of these isolates from water and from fish is shown in Appendix A.

*Saprolegnia* Species	Water	Fish	Total
*S. australis*	8	6	14
*S. delica*	5	5	10
*S. diclina*	12	8	20
*S. ferax*	10	23	33
*S. parasitica*	21	130	151
*S. parasitica* phylotypes			
S2	7	100	107
S3	1	1	2
S4	0	2	2
S5	0	4	4
S6	13	23	36

**Table 4 jof-10-00057-t004:** Total number of isolates of each *Saprolegnia* species obtained from enzootic and epizootic periods in Atlantic salmon farms in Scotland. A phylogenetic analysis of these isolates from water and from fish is shown in Appendix A.

*Saprolegnia* Species	Enzootic	Epizootic	Total
*S. australis*	8	6	14
*S. delica*	4	6	10
*S. diclina*	16	4	20
*S. ferax*	33	0	33
*S. parasitica*	103	45	148
*S. parasitica* phylotypes			
S2	65	41	106
S3	0	3	3
S4	2	0	2
S5	0	1	1
S6	36	0	36

## Data Availability

The nucleotide sequence of the ITS region for all 412 isolates from the Scottish Atlantic salmon aquaculture sites are deposited in GenBank (accession numbers OQ6784-08 to OQ6787-89).

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
