# Peer review of "Specific Phylotypes of Saprolegnia parasitica Associated with Atlantic Salmon Freshwater Aquaculture"

_jof, 2024, doi:10.3390/jof10010057_

Round 1

Reviewer 1 Report

Comments and Suggestions for Authors

I enjoyed reading the manuscript, and I appreciate the effort that went into it. The phylogenetic analyses are thorough. However, I have some suggestions for improving the presentation of the results.

Major Comments:

1. Phylotypes based on ITS have relatively low resolution. It might be worth considering alternative approaches. For example, a recent study by Ravasi et al. (2018) introduced an S. parasitica MLST typing scheme, which assigned 10 different genotypes to the dataset of isolates from Switzerland. In contrast, the ITS-based approach only distinguished 3 S. parasitica ITS-phylotypes in the same dataset (as far as I can tell from their ITS tree). It would be helpful to explore how these DSTs relate to the phylotypes. However, I understand that sequencing the ITS region is faster, and there is an abundance of ITS sequences already available in databases for comparison. So, I'm fine with the current approach.

2. It appears that only Saprolegnia isolates were identified using both the best BLAST hit method and phylogenetic analyses with reference sequences from the literature. Were these two identification methods consistent for all Saprolegnia isolates? As noted by Sandoval-Sierra et al. (2014), GenBank is known for errors in species identification due to misassigned sequences and sequencing errors.

3. Isolates from other genera were identified solely using the best BLAST hit approach. I'm curious about the reliability of this method in all cases. Can you provide more details? For example, you mentioned, "Of the three most abundantly isolated Pythium species, only P. coloratum and P. flevoense could be identified with confidence via NCBI BLAST." What does "with confidence" mean here? Clarification is needed. I think appropriate reference sequences from the literature for other genera must be included in the tree presented in Figure 1, even though I understand that the manuscript primarily focuses on the genus Saprolegnia. This would help avoid introducing misassigned sequences into GenBank.

4. When comparing isolates based on their source of isolation (water/fish; endozootic/epizootic), the data is presented in tabular form and as two separate trees for each comparison. This presentation of isolates from different sources in different trees doesn’t make sense to me. I would delete Figures 4 and 5. Instead, I suggest conducting statistical comparisons between the sampling sources (see for instance the approach used in Sandoval-Sierra et al., 2014). Additionally, it would be helpful to include a table or statistical comparison of isolates from recirculating vs. flow-through sites, as this is a recurring point in the discussion.

5. The manuscript provides a detailed and interesting discussion of the geographical distribution of different S. parasitica isolates. It seems the authors have checked for the presence of phylotypes in the databases. I recommend presenting this information in a table, either within the results section or the discussion. This table should include all relevant references (4, 14, 15, 30, 31, 32, 33, and Pavić et al., 2021), along with the respective countries and, if available, host species. In this way, the currently known distribution of different S. parasitica phylotypes will be easier to follow.

Minor Comments:

Abstract:

Lines 18 and 19: Delete the part of the sentence that mentions making separate trees based on the origin of isolates. See also my comment above.

Introduction:

Line 68: Italicize "Saprolegnia" (and apply italics in other relevant places in the manuscript).

Materials and Methods:

Line 43: Remove the word "alignment”.

Line 51: Provide a brief explanation of the "Z3 epizootic identification model" for the benefit of readers who may not be familiar with it (such as myself – I don’t understand this).

Line 58: Use superscript for "106."

Reviewer 2 Report

Comments and Suggestions for Authors

The manuscript jof-2682412 describes the diversity of Saprolegnia species in the waters and aquacultured salmon of some Scottish sites.

The authors describe all activities flawlessly. What else can I say: excellent work.

Just to justify the reviewer's work, we point out:

- line 445 insert the dotted S at the end of the line;

- line 526 remove the closing parenthesis after the citation;

- line 530 write Saprolegnia in italics.

This manuscript can be submitted for publication after minor revision.

Round 2

Reviewer 1 Report

Comments and Suggestions for Authors

The authors have improved the manuscript. I have only minor comments left.

Could you add the protocol for identification of non-Saprolegnia isolates to the manuscript? The protocol described by the authors in their reply to my previous comments makes sense and I think it would be beneficial to readers of the manuscript.

I still don’t see the point of leaving Figures 4 and 5 in the manuscript. Even without the statistical comparison, the differences in the composition of the phylotypes from water and fish are clearly visible in Table 3.

Lines 149-152: I suppose a threshold is missing. Please add.

Line 159: x 106

Lines 486-488: What about the phylotypes of the remaining 36% of isolates?

Line 501: Esox lucius should be in italics.

Line 517: Correct to Saprolegniaceae.

Line 523: “has only been previously reported in Canada”.

Line 528: Delete “and”.

Line 584: Dot missing.

Table 5: Please sort the table by country, then by phylotype.

Author Response

Comments and Suggestions for Authors

The authors have improved the manuscript. I have only minor comments left.

Response: We appreciate all comments as it has strengthened the manuscript.

Could you add the protocol for identification of non-Saprolegnia isolates to the manuscript? The protocol described by the authors in their reply to my previous comments makes sense and I think it would be beneficial to readers of the manuscript.

Response- We have now added the protocol to manuscript

I still don’t see the point of leaving Figures 4 and 5 in the manuscript. Even without the statistical comparison, the differences in the composition of the phylotypes from water and fish are clearly visible in Table 3.

Response – We have removed Figures 4 and 5 and placed them in supplementary data. We hope that is acceptable.

Lines 149-152: I suppose a threshold is missing. Please add.

Response -We are not sure what is required since (see below) the first sentence is ‘default’. The second is ‘identical’ so 100%. The third does not really have a threshold.

Regarding: ‘All alignments were performed using MUSCLE [21] with default settings and a maximum of 8 iterations. Consensus sequences were produced for all isolates with identical ITS sequences and used as a representative sequence for the respective species/phylotype. Alignment trimming was performed in Geneious Prime v2020.1.2’.

Line 159: x 106            DONE

Lines 486-488: What about the phylotypes of the remaining 36% of isolates?

Response – This refers to 64% quoted as S2 isolates from paper Pavic et al, however this should be 86%. An additional 22% has been added as we stated in the Table that we assigned S2 using a cut off of 600bp ITS.

Regarding the remaining 14%, we feel that this is not relevant for our paper but we know that 4 isolates (8%) had single change at 414 not observed in designated phylotypes S1 – S6, that could be designated another phylotype (out of scope of paper) – this change seems unique to these Croatian isolates. The remaining 3 isolates (6%) had 3, 11, 29 different changes.

Line 501: Esox lucius should be in italics.                                   DONE

Line 517: Correct to Saprolegniaceae.                                       DONE

Line 523: “has only been previously reported in Canada”.        DONE

Line 528: Delete “and”.                                                              DONE

Line 584: Dot missing.                                                               DONE

Table 5: Please sort the table by country, then by phylotype.  DONE

Round 3

Reviewer 1 Report

Comments and Suggestions for Authors

I am satisfied with the authors' answers and corrections and think that the manuscript can be accepted.

Regarding the missing threshold, I referred to the z-score model (but obviously wrote the wrong line numbers). It is lines 157-160.